# RNA Modifications in Pathogenic Viruses: Existence, Mechanism, and Impacts

**DOI:** 10.3390/microorganisms12112373

**Published:** 2024-11-20

**Authors:** Yingying Zou, Zhoule Guo, Xing-Yi Ge, Ye Qiu

**Affiliations:** Hunan Provincial Key Laboratory of Medical Virology, College of Biology, Hunan University, Changsha 410012, China; yyzou@hnu.edu.cn (Y.Z.); guozhoule@hnu.edu.cn (Z.G.)

**Keywords:** RNA modification, viral RNA, N6-methyladenosine, 5-methylcytosine, N4-acetylcytosine, N1-methyladenosine

## Abstract

RNA modification is a key posttranscriptional process playing various biological roles, and one which has been reported to exist extensively in cellular RNAs. Interestingly, recent studies have shown that viral RNAs also contain a variety of RNA modifications, which are regulated dynamically by host modification machinery and play critical roles in different stages of the viral life cycle. In this review, we summarize the reports of four typical modifications reported on viral RNAs, including N6-methyladenosine (m6A), 5-methylcytosine (m5C), N4-acetylcytosine (ac4C), and N1-methyladenosine (m1A), describe the molecular mechanisms of these modification processes, and illustrate their impacts on viral replication, pathogenicity, and innate immune responses. Notably, we find that RNA modifications in different viruses share some common features and mechanisms in their generation, regulation, and function, highlighting the potential for viral RNA modifications and the related host machinery to serve as the targets or bases for the development of antiviral therapeutics and vaccines.

## 1. Introduction

Chemical modifications of ribonucleotides are extensively distributed in coding and non-coding RNAs, and are dynamically regulated by proteins specifically responsible for adding, removing, and recognizing the modifications [1]. RNA modifications affect RNA biogenesis, localization, transport, splicing, decay, and translation, and therefore have a non-negligible impact on RNA metabolism and function [2]. More importantly, RNA modifications play indispensable roles in many biological processes as regulators of gene expression. Abnormal RNA modifications may lead to diseases, such as cancer, metabolic diseases, and immune-related diseases, and drug development targeting RNA modifications is an emerging direction for the treatment of related diseases [3]. In addition to modifications in cellular RNA, viruses have also been found to utilize the cellular modification machinery to add covalent modifications to their own RNA [4,5].

As epigenetic sequencing technologies continue to mature, much progress has been made in understanding the existence and functions of RNA modifications in viruses, especially in recent years. In this review, we summarize four typical RNA modifications that have been reported in viral RNA, including N6-methyladenosine (m6A), 5-methylcytosine (m5C), N4-acetylcytosine (ac4C), and N1-methyladenosine (m1A), especially in the context of their roles in viral replication, pathogenesis, and antiviral immunity. In addition, we discuss the potential application of RNA modifications in antiviral prevention and therapy development.

## 2. Mechanism of RNA Modifications

### 2.1. N6-methyladenosine (m6A)

The methylation of adenosine nitrogen position 6 (m6A), is the most abundant internal modification in eukaryotic RNAs. Three groups of cellular proteins are involved in the process of m6A modification: (1) adenosine methyltransferases (known as writers), which catalyze the formation of m6A in RNA; (2) adenosine demethylases (known as erasers), which mediate the removal of m6A from RNA; and (3) m6A-binding proteins (known as readers), which recognize nucleotides undergoing m6A modification to drive downstream signals. Methyltransferase-like (METTL) proteins METTL3 and METTL14 are typical m6A writers which have been reported recently; they function in complexes with their cofactors, such as Wilms tumor 1-associating protein (WTAP), Vir-like m6A methyltransferase-associated (VIRMA), Cbl proto-oncogene like 1 (HAKAI), zinc finger CCCH-type containing 13 (ZC3H13), and RNA-binding motif 15/15B (RBM15/15B). METTL3-mediated m6A addition usually occurs at the consensus motif DRAmCH in messenger RNA (mRNA), where D = G/A/U, R = G > A, and H = U/C/A [6,7,8,9]. In addition, METTL5, METTL16, and zinc finger CCHC-type containing 4 (ZCCHC4) have been found to be responsible for m6A addition in ribosomal RNA (rRNA) and small nuclear RNA (snRNA) [10,11,12]. Only two m6A erasers have been reported to date, namely, the fat mass and obesity-associated (FTO) protein and AlkB Homolog 5 (ALKBH5) [13,14]. The major m6A readers identified include the cytoplasmic proteins YTH domain family 1-3 (YTHDF1-3) and YTH domain containing 2 (YTHDC2), as well as the nuclear protein YTHDC1; these bind to m6A through its C-terminal YTH structural domain and thus mediate their specific biological functions. YTHDF1 preferentially binds m6A in the 3′ untranslated region (3′ UTR) and promotes translation by interacting with translation initiation factors, whereas YTHDF2 recruits the carbon catabolite repression 4 (CCR4)-negative on the TATA-less (NOT) complex through its N-terminal structural domain to promote degradation of m6A-modified mRNAs. YTHDF3 affects the translation and decay of methylated mRNAs by collaborating with YTHDF1 and YTHDF2 [15,16]. YTHDC1 regulates mRNA splicing and nuclear export, while YTHDC2 contributes to RNA degradation and translation [17]. Other readers have also been discovered, including some heterogeneous nuclear ribonucleoproteins (HNRNPs) and several insulin-like growth factor 2 mRNA-binding proteins (IGF2BPs), which are important for RNA splicing and RNA stabilization, respectively [18,19]. The discovery of these three groups of proteins implies that the biological processes involved by m6A modifications are dynamic and reversible.

### 2.2. 5-methylcytosine (m5C)

The methylation of cytosine residue position 5 (m5C), is another modification widely found in eukaryotic RNAs [20]. Members of the NOL1/NOP2/SUN domain (NSUN) family (NSUN1-NSUN7) are responsible for catalyzing m5C modifications, among which NSUN2 has the broadest substrate specificity [21,22]. In addition, DNA methyltransferase homolog DNMT2 is also found to mediate the formation of m5C in RNAs [21,23]. The m5C demethylation can be catalyzed by ten-11 translocation 2 (TET2) [24]. Aly/REF export factor (ALYREF) and Y-box binding protein 1 (YBX1) have been proved to be m5C readers that recognize and bind m5C-modified mRNAs to promote nuclear export and the stability of transcripts, respectively [25,26].

### 2.3. N4-acetylcytosine (ac4C)

The ac4C modification, the only acetylation modification found in RNAs of eukaryotic cells and viruses infecting eukaryotic cells, occurs at the N4 site of the cytosine base. N-acetyltransferase 10 (NAT10) and its homologs are writers responsible for the addition of ac4C on different RNA substrates [27]. The nuclear protein NAT10 possesses both acetyltransferase activity and RNA-binding activity, and it uses acetyl coenzyme A as an acetyl donor to catalyze the intermediate cytosine acetylation of most 5′-CCG-3′ consensus motifs in RNAs driven by ATP hydrolysis [28]. Currently, only one NAD^+^-dependent protein deacetylase, sirtuin 7 (SIRT7), has been identified as an ac4C eraser for rRNA [29]. Other RNA deacetylases and ac4C readers still remain to be discovered. The ac4C modifications were initially discovered in transfer RNAs (tRNAs) and rRNAs, and promote tRNA stability and correct codon recognition [30], as well as aiding in rRNA processing and ribosome biogenesis [31]. Later, ac4C was also found to be an mRNA modification that promotes mRNA stability and translation efficiency [32].

### 2.4. N1-methyladenosine (m1A)

The N1-site adenosine methylation modification m1A is widely present in different RNAs, mostly found in tRNAs. The m1A writers include the tRNA methyltransferases TRMT6, TRMT61A, TRMT10C, and TRMT61B [33]. Among them, TRMT61B and TRMT10C are responsible for m1A formation on mitochondrial tRNAs or rRNAs, whereas the complex formed by a catalytic subunit TRMT61A and an RNA-binding subunit TRMT6 is responsible for the formation of m1A on cytoplasmic tRNAs and mRNAs with GUUCRA tRNA-like motifs. In addition, a variety of m1A demethylases, including ALKBH1, ALKBH3, ALKBH7, and FTO, and m1A-binding proteins, including YTHDF1, YTHDF2, YTHDF3, and YTHDC1, have been identified [34]. The m1A has been found to primarily affect RNA structure stabilization and protein synthesis [3,35].

## 3. Methods of Viral RNA Modifications Detection

In recent years, an increasing number of viral RNA modifications have been reported in a variety of pathogenic viruses, and the impacts of these modifications on the viral life cycle and viral pathogenicity have been attracting increased attention. The loci where viral RNA modifications occur are important for studying their specific functions, and hence a significant number of techniques for viral RNA modification detection have been developed. 

Liquid chromatography tandem mass spectrometry (LC-MS/MS) is used to quantify the modifications in purified RNAs extracted from released viral particles or virus-infected cells, although this method can only detect RNA modifications at a low resolution, and without the accurate determination of modification sites [36,37]. 

Advances in high-throughput sequencing technologies have promoted the high-resolution detection of the distribution and abundance of viral RNA modifications [38,39]. Currently, a central strategy for the identification of viral RNA modification sites is next-generation sequencing (NGS), following immunoprecipitation using modification-specific antibodies. For instance, RNA-immunoprecipitation sequencing (RIP-seq) is capable of providing profiles of several different viral RNA modifications, typically those for m6A, m5C, ac4C, and m1A, rendered at a resolution of 100–200 nucleotides, by slicing RNAs to uniform fragments and then enriching modified RNA fragments using specific antibodies [37,40,41,42]. For further improvement, photo-crosslinking-assisted (PA) m6A/m5C/ac4C sequencing (PA-m6A/m5C/ac4C-seq) [43,44,45,46] and m6A individual-nucleotide-resolution crosslinking and immunoprecipitation sequencing (miCLIP-seq) [47,48] allow for higher, and even single-base, resolution detection of viral RNA modifications by introducing ultraviolet light-induced cross-linking of antibodies to modified RNAs to generate mutations or truncations at modification sites during reverse transcription. Moreover, crosslinking and immunoprecipitation (CLIP) and photoactivatable-ribonucleoside-enhanced CLIP (PAR-CLIP) sequencing are also used to probe binding sites on viral RNA for modification machinery proteins [45,49,50]. Although these antibody-based mapping methods are fast and can be used for the detection of modifications in low-abundance RNAs, they still face some problems, such as affinity variation and cross-reactivity, which might cause false positives [39]. To circumvent antibodies, bisulfite sequencing (BS-seq) and sodium borohydride (NaBH_4_) sequencing (RedaC:T-seq), two typical antibody-independent techniques based on biochemical mutation characterization, have been developed to detect m5C and ac4C at single-nucleotide resolution, respectively [51,52,53,54]. However, the accuracy of BS-seq is often affected by incomplete C-to-U conversion due to RNA secondary structure, a factor which requires optimization of the experimental conditions [55]. In comparison, the main limitation of RedaC:T-seq is off-target deacetylation in basic reaction conditions, which imposes restrictions on the efficiency of NaBH_4_-mediated ac4C reduction [56]. 

Recently, nanopore direct-RNA sequencing technology (DRS) has provided a powerful tool for identifying viral RNA modifications at a single-nucleotide resolution [57,58,59]. DRS can directly detect modified nucleotides in RNA by means of a generated characteristic-current blockade within the nanopore, thus avoiding the bias caused by reverse transcription and cDNA amplification. Moreover, this method can read sequences much longer than those readable by NGS, which is beneficial for analyzing alternative splicing variants, though the sequencing accuracy of this method needs to be improved [60]. 

In the future, more methods with higher sensitivities and specificity are expected to be applied to characterize viral RNA modifications, such as the chemically labeled m6A sequencing technologies (e.g., m6A-SEAL [61] and m6A-label-seq [62]), deamination adjacent to RNA modification targets (DART-seq) [63] enabling m6A detection at the single-cell level, glyoxal and nitrite-mediated deamination of unmethylated adenosines (GLORI) [64] which enables absolute quantification of m6A, and m1A-quant-seq [65] for estimating individual-site m1A stoichiometry.

## 4. Roles of m6A in Viral RNAs

The discovery of m6A in viral RNAs can be traced back to the 1970s [66,67,68], but due to technological limitations, its function was not characterized for a long time. In recent years, based on the development of epitranscriptome sequencing technologies, the sites of m6A modification in many viral RNAs have been determined, and their roles have also been well studied (Table 1).

### 4.1. HIV-1

Human immunodeficiency virus (HIV), including HIV-1 and HIV-2, is a member of the *Lentivirus* genus in the *Retroviridae* family, and has two identical positive-stranded genomic RNAs. When HIV enters immune cells, its genetic materials can be synthesized into double-stranded DNA molecules through reverse transcription, and can then be integrated into the host-cell DNA and transcribed into new viral RNAs. HIV can destroy the host immune system, leading to acquired immunodeficiency syndrome (AIDS), of which HIV-1 is the predominant strain worldwide [69].

Recent studies have mapped m6A modifications in HIV-1 RNAs using MeRIP-seq and PA-m6A-seq, with most m6A sites overlapped with YTHDFs and YTHDC1-binding sites shown by PAR-CLIP [50,70,71,72,73]. In addition, Tirumuru et al. quantified approximately 3–4 m6A bases in each HIV-1 genomic RNA (gRNA) copy using LC-MS/MS [50]. However, the locations and abundance of the m6A distribution in these reported HIV-1 RNAs are different. In particular, Kennedy et al. identified several m6A peaks which were located only in the 3′ UTR of the viral genome, whereas several other studies also identified m6A peaks in the 5′ UTR and coding sequence (CDS) [50,70,71,72]. This discrepancy may be due to differences in viral strains, infected cell types, and identification methodologies.

The m6A modifications, as regulated by three machinery proteins, play complex roles at different stages of the HIV-1 life cycle. METTL3/METTL14-catalyzed m6A modifications in HIV-1 Rev response element (RRE) RNA, especially methylation of the conserved site A7883, enhance the binding of Rev proteins to the RRE, which helps viral RNA nuclear export and viral replication [70]. In contrast, AlkBH5 and FTO negatively regulate HIV-1 replication and virus release by reducing m6A levels [50,70,73]. The m6A sites in the 3′ UTR of HIV-1 gRNA recruit YTHDFs to promote viral mRNA expression [71]. Direct mutation of two potential m6A sites in the 5′ UTR of HIV-1 gRNA significantly reduces viral infectivity [74]. These precise results emphasize the positive regulatory effect of m6A on HIV-1 infection. However, it has also been reported that the METTL3/METTL14 complex, which increases m6A deposition in HIV-1 RNAs, acts as a negative regulator of viral full-length RNA packaging and HIV-1 mRNA levels, while FTO promotes viral RNA packaging and particle release [72,75]. Similarly, m6A readers perform dual functions in the HIV-1 life cycle, which may be related to their high affinity to m6A-modified HIV-1 gRNAs. During the early stage of the HIV-1 replication cycle, the YTHDF1-3 inhibits viral reverse transcription and negatively regulates subsequent viral gene expression and HIV-1 infection [50,74,76], but in HIV-1-producing cells, where the virus has been integrated into the cell, the YTHDF1-3 regulates viral protein expression and particle release positively [74]. Of special note, Jurczyszak et al. found that YTHDF3 was an anti-HIV protein incorporated into released viral particles through interaction with a Gag nucleocapsid protein that inhibited viral reverse transcription in the subsequent round of infection, but also that the virus cleaved and inactivated YTHDF3 by viral proteases to antagonize this process [76]. However, this finding is still controversial, since other studies have not detected any m6A readers in mature HIV-1 particles [73,74]. In addition, YTHDC1 can bind to m6A-modified HIV-1 RNAs and it regulates the selective splicing of viral RNAs without affecting RNA nuclear export or stability. Similar to YTHDFs, YTHDC1 appears to differentially regulate distinct life-cycle phases in HIV-infected and HIV-produced cells [73,75].

Interestingly, m6A-deficient HIV-1 RNAs can be recognized by the cytoplasmic RNA sensor, retinoic acid inducible gene I (RIG-I), which promotes type I interferon (IFN-I) expression via activation of interferon regulatory factors IRF3 and IRF7, suggesting the important role of m6A modifications in HIV-1 RNAs in viral evasion from the host innate immune system [77].

### 4.2. SARS-CoV-2

Severe acute respiratory syndrome coronavirus clade 2 (SARS-CoV-2) belongs to the *Betacoronavirus* genus of the *Coronaviridae* family and has a length of approximately 30 kb, comprising single-stranded positive-sense genomic RNA [78]. SARS-CoV-2 is the causative agent of coronavirus disease 2019 (COVID-19), the cause of pandemics since its outbreak in 2019, and one which has posed a major threat to global public health [79].

Most studies have shown that m6A modifications are more clustered in the 3′ end of the SARS-CoV-2 genome, especially in the *N* region, though differences in the infected cell types, infection times, and assay protocols may affect the location and amount of m6A deposited in the SARS-CoV-2 RNA [37,47,57,80,81,82]. Knockdown or activity-inhibition of METTL3, which reduces the abundance of m6A in SARS-CoV-2 RNAs, was detrimental to SARS-CoV-2 replication and pathogenicity [19,37,57,80,83]. Specifically, the RNA-dependent RNA polymerase (RdRp) of SARS-CoV-2 interacts with METTL3 to promote the latter’s protein expression and methylating activity, possibly by inhibiting METTL3 sumoylation and K48/63-linked ubiquitination [57]. Conversely, knockdown of FTO, the m6A eraser, promoted viral replication [57]. As for m6A readers, three YTHDFs positively regulate SARS-CoV-2 infection [80]. Kumar et al. reported that another m6A reader, HNRNPA1, was recruited into m6A-modified SARS-CoV-2 RNA to promote viral RNA synthesis without affecting viral attachment, entry, and release [19]. These results suggest a positive regulation of the SARS-CoV-2 viral life cycle by means of m6A modifications. However, another study reported that METTL3, METTL14, and YTHDF2 inhibited viral infection, replication, and release, indicating the potential negative effects of m6A on SARS-CoV-2 [47]. Interestingly, Becker et al. identified an m6A site (m6A74) located on the short stem-loop structure SL3 in the 5′ UTR of SARS-CoV-2 gRNA. The m6A74 can reduce the stability of the 5′ SL3 stem-loop without affecting its secondary and 3D structures. This site is located on the viral transcriptional regulatory leader sequence (TRS-L) and destabilizes the binding between the positive-stranded TRS-L and the negative-stranded TRS-B sequences of four viral structural proteins, blocking discontinuous transcription of viral subgenomic mRNAs. Also, m6A74 modification alters the equilibrium between the 5′ SL3 hairpin and the TRS duplex, which, in turn, can affect viral structural protein synthesis [84].

Moreover, m6A modifications in the *N* region of SARS-CoV-2 RNA play an important role in the immune evasion of the virus. Knockdown of METTL3 or mutation of m6A sites in the viral *N* region results in increased binding of RIG-1 to the *N* region and facilitates the activation of the downstream innate immune signaling pathway and the expression of inflammatory genes [37].

Although no experiments have explored the effects of m6A modifications in viral RNA on pathogenicity, bioinformatic analyses have shown that m6A in SARS-CoV-2 positive-stranded gRNAs, but not negative-stranded RNAs, correlate with viral pathogenicity. In addition, two predicted m6A sites associated with severe COVID-19 (located in the 5′ UTR and S-ORF3a) and five predicted m6A sites associated with mild COVID-19 (located in ORF1ab and N) were identified on the Delta lineage sequences of infected patients [83].

### 4.3. IAV

Influenza A virus (IAV) is an enveloped virus belonging to the *Orthomyxoviridae* family. Its genome contains eight single-stranded negative-sense RNA fragments. IAV expresses two surface glycoproteins, hemagglutinin (HA) and neuraminidase (NA). Currently, 18 HA variants (H1-H18) and 11 NA variants (N1-N11) have been identified. IAV is capable of obtaining different combinations of HA and NA via reassortment, generating many subtypes. IAV infects a wide range of hosts and cross-species transmission of IAV is the principal cause of influenza pandemics [85].

The m6A modification was first detected in IAV (WSN/H1N1) mRNAs in 1976 [67]. Subsequently, 24 m6A residues in IAV mRNAs were identified by biochemical analyses, and the numbers and the patterns of their distribution in different mRNAs were described [67,86]. However, the specific sites and functions of m6A in IAV RNAs remain unclear.

In recent years, m6A sites have been mapped on the positive-strand mRNAs and the negative-strand viral RNAs (vRNAs) of IAV (PR8/H1N1) by PA-m6A-seq, and these sites are highly consistent with the YTHDF binding sites shown by PAR-CLIP [43]. Interestingly, m6A modifications are distributed at higher levels on RNAs encoding the viral structural proteins HA, NP, NA, and M (M1/M2) than on RNAs encoding the three viral RdRp subunits, PB2, PB1, and PA. In particular, eight and nine major m6A peaks are identified in HA mRNA and vRNA, respectively. Synonymous mutations of the 5′-RAC-3′ consensus motif corresponding to these sites result in suppression of HA mRNA and protein levels. In addition, two IAV strains with a mutant m6A site in their positive- or negative-strand RNAs also showed a significant reduction in pathogenicity when infecting mice, suggesting a positive regulatory effect of m6A modification on IAV replication and pathogenicity. Notably, fourteen 5′-DRACH-3′ motifs in HA mRNAs are partially conserved across all HA subtypes (H1-18), with six of them highly conserved across all H1 subtypes. However, the total number of DRACH motifs and conserved motifs are lower in HA vRNAs compared to mRNAs, suggesting a conserved function of m6A modifications on mRNA translation and stability across different subtypes and strains of IAVs [87]. Another conserved m6A site, G/GA(530)C, overlaps with the 3′ splicing site (CAG/GAC) in IAV NS mRNA. YTHDC1 recognizes and binds to this site to inhibit NS mRNA splicing [88]. Overall, m6A modifications benefit IAV replication and pathogenicity, and this function may be conserved.

### 4.4. Flavivirus

Flaviviruses are enveloped, single-stranded, positive-sense RNA viruses belonging to *Flaviviridae* family, and which are further classified into four genera: *Flavivirus*, *Hepacivirus*, *Pegivirus*, and *Pestivirus* [89]. Of these, members of the *Flavivirus* and *Hepacivirus* genera pose a serious health risk to humans. Infection with Zika virus (ZIKV), dengue virus (DENV), West Nile virus (WNV) and yellow fever virus (YFV) in the *Flavivirus* genus can cause febrile illness, severe hemorrhage, shock, and neuroinflammation. ZIKV infection has also been associated with a variety of neurodevelopmental defects. Infection with hepatitis C virus (HCV), a member of the *Hepatovirus* genus, puts patients at risk for liver fibrosis, liver damage, cirrhosis, and liver cancer [90].

The m6A nucleosides are abundant in flavivirus RNAs, including HCV, ZIKV, DENV, YFV, and WNV. Overall, m6A modifications are distributed over the coding regions of non-structural proteins NS3 and NS5 (NS5A and NS5B for HCV) of all these viruses. In addition, most of the m6A consensus motifs in the 12 m6A peaks identified in ZIKV RNAs are consistent among different ZIKV strains, suggesting a conserved role of this post-transcriptional modification in viral replication [36,91]. The m6A modifications regulate the life cycles of flaviviruses. METTL3 and METTL14, as well as ALKBH5 and FTO, affect viral replication and particle production by controlling the addition and removal of m6A in ZIKV RNAs, respectively; YTHDF1-3 binds to m6A-modified ZIKV RNAs, which negatively regulates viral replication [36]. In addition, METTL3 and METTL14 negatively regulate HCV RNA stability, protein expression, and viral particle release [91,92,93]. YTHDFs can bind the m6A sites on the *E1* region of the HCV genome, which encodes viral envelope glycoprotein, to reduce the interaction between viral RNA and core proteins, thereby inhibiting the assembly and release of infectious viral particles [91]. Nevertheless, a different series of m6A modifications may benefit flavivirus infection. For instance, some m6A modifications in ZIKV RNA may promote viral replication by decreasing mitochondrial antiviral signaling protein (MAVS)-mediated interferon response [94]. In addition, Kim et al. identified an m6A site in the internal ribosome entry site (IRES) of the HCV genome (A331), which is recognized by YTHDC2 to enhance HCV translation initiation, dependent on its helicase activity and interaction with cellular La antigen [92]. Furthermore, a specific m6A modification is identified at nucleotide (nt) 8766 of HCV genomic RNA, which is about 100 nt upstream of the pathogen-associated molecular pattern (PAMP) RNA recognized by RIG-I. YTHDF2 binds to this m6A site to assist viral evasion of the RIG-I-mediated I-IFN signaling response [95]. In conclusion, m6A modifications play different functions in the life cycles of flavivirus, especially as reported in ZIKV and HCV, depending on their locations in the viral RNAs.

### 4.5. EV71

Enterovirus 71 (EV71) belongs to the *Enterovirus* genus of the *Picornaviridae* family and has a non-enveloped, single-stranded, positive-sense RNA genome. EV71 is the main pathogen causing hand, foot, and mouth disease (HFMD), and also causes various neurological complications, which seriously threatens public health [96].

MeRIP-seq data from human and monkey cell lines show that relatively conserved m6A modifications regulated by METTL3 and FTO are deposited on the VP4, VP1, and 2C coding regions of the EV71 RNA genome [97,98]. Hao et al. identified two m6A sites, A3055 and A4555, on the *VP1* and *2C* regions, by primer extension analysis. Interestingly, the mutation of the cytidine residue of certain GAC motifs inhibits m6A addition and leads to a decrease in progeny viral production [97]. Yao et al. reported the promotive effect of m6A modifications in the EV71 5′ UTR-*VP4* junction on viral translation [98]. Knockdown of METTL3 reduces EV71 RNA copies, VP1 protein expression, and progeny virus production, while knockdown of FTO promotes viral RNA replication [97,98,99]. In RD cells, YTHDF1-3 and YTHDC1 proteins negatively regulate viral genome replication [97]. The m6A machinery proteins may affect the EV71 life cycle by regulating m6A in viral RNAs. In conclusion, m6A modifications tend to promote EV71 replication.

### 4.6. Pneumovirus

Pneumoviruses are single-stranded, negative-sense, enveloped RNA viruses belonging to the *Pneumovirinae* subfamily of the *Paramyxoviridae* family [100]. Among them, respiratory syncytial virus (RSV), of the *Orthopneumovirus* genus, and human metapneumovirus (hMPV), of the *Metapneumovirus* genus, are common pathogens of human respiratory infections [101,102]. RSV and hMPV infections can cause similar symptoms such as severe cough, fine bronchitis, pneumonia, and even death [100,103].

The m6A-seq data revealed prominent and overlapping m6A modification signals in the *G* region of the genomes and anti-genomes for RSV and hMPV [104,105]. Synonymous mutation of these potential m6A sites promoted viral RNA binding to RIG-I and activated RIG-I expression to help induce downstream IFN-I responses, which reduced replicative capacity and activity of m6A-deficient RSV and hMPV [104,105,106]. In addition, m6A writers (METTL3, METTL14) and readers (YTHDF1-3) positively regulate RSV and hMPV replication [104,105]. However, YTHDC1 exerts different functions on these two viruses, in that it promotes hMPV viral replication but negatively regulates RSV infection [105,107]. In summary, m6A modifications in RSV and hMPV RNAs mainly help the viruses evade RIG-I-mediated innate immunity, thereby promoting viral replication.

### 4.7. VSV

Vesicular stomatatis virus (VSV) belongs to the *Vesiculovirus* genus and the *Rhabdoviridae* family, and has a negative-sense, single-stranded RNA genome [108]. VSV infects various natural hosts including humans, causing flu-like symptoms and blister-like lesions [109].

Currently, only one study has reported the deposition of m6A on the 3′ and 5′ terminals of VSV N, P, M, and G positive-sense RNAs, with no m6A being detected in viral negative-sense gRNA [110]. METTL3-mediated m6A sites on viral transcripts can inhibit viral double-stranded RNA (dsRNA) formation and thereby suppress the recognition of RIG-I and melanoma differentiation-associated gene 5 (MDA5), leading to evasion from innate immunity. More importantly, METTL3 deficiency enhanced IFN-β-dependent innate immune responses and suppressed viral replication and pathogenicity in mice, suggesting METTL3 as an antiviral therapeutic target.

### 4.8. HBV

Hepatitis B virus (HBV) belongs to the *Orthohepadnavirus* genus and the *Hepadnaviridae* family, and has a partial double-stranded DNA (rcDNA) genome. The rcDNA forms covalently closed DNA (cccDNA) in the nucleus, which serves as the template for transcription of viral mRNAs, encoding viral proteins including HBs, HBc, HBe, HBx, and polymerase (Pol). HBV replicates using pre-genomic RNAs (pgRNAs). The pgRNAs are packaged as nuclear capsids in the cytoplasm and subsequently produce rcDNAs by reverse transcription to generate mature core particles which can be released to the extracellular compartment as infectious viral particles or translocated to the nucleus for conversion into cccDNA [111,112]. HBV infection can lead to serious diseases such as hepatitis, cirrhosis, and hepatocellular carcinoma [113].

The m6A modifications in HBV transcripts exert various functions in different life-cycle stages of the virus. Most studies have focused on m6A modifications in the epsilon stem loop of HBV pgRNAs and in the HBx coding region, which are critical for viral transcription and replication [111,114]. The conserved m6A at A1907 in the 3′ epsilon stem-loop of pgRNA negatively regulates HBV RNA stability and viral protein expression, whereas m6A1907 in the 5′ epsilon stem loop aids nuclear export and reverse transcription of HBV transcripts, as well as cccDNA synthesis [115,116,117]. METTL3/METTL14, YTHDF2, and YTHDC1 play similar regulatory roles by affecting m6A addition and recognition [115,116,117,118]. Moreover, m6A modifications in epsilon stem loop affect innate immune response, though with ambiguity as to their specific function. The m6A1907 in the 5′ and 3′ epsilon stem loops was suggested by Imam et al. as a recognition marker for IFN-α-induced interferon-stimulated gene (ISG) 20, which mediated the degradation of HBV RNA, and this process was dependent on YTHDF2 and METTL3/METTL14 [119]. However, Kim et al. found that YTHDF2 bound 5′ epsilon m6A-modified pgRNA to evade RIG-I recognition and inhibit IRF3 phosphorylation [95]. The m6A modification at A1616 in the viral HBx coding region specifically inhibits HBx protein expression by interacting with YTHDF2, whereas the other m6A sites (A1662, A1670, A1714, and A1729) synergistically positively regulate HBV mRNA and HBs protein levels [118,120]. Remarkably, viral protein HBx mutually interacts with host m6A machinery proteins to regulate viral infection. During HBV infection, viral HBx protein interacts with METTL3/14 complex and facilitates its entry into the nucleus to support m6A addition on HBV transcripts, thereby negatively regulating the stability and translation activity of viral transcripts [121]. Additionally, HBx protein increases ALKBH5 expression by promoting the H3K4me3 modification of *ALKBH5* gene promoter, which may in turn benefit HBx mRNA stability due to decreased m6A levels on the 3′ UTR of the HBx mRNA [122].

In addition to m6A in HBV RNAs, a large number of articles have reported that m6A machinery proteins also regulate host gene expression during HBV infection to influence hepatocellular carcinoma (HCC) development, emphasizing the value of m6A modification in antiviral or anticancer applications [123,124]. Furthermore, bioinformatic analysis has revealed that m6A regulators can be used as targets for HBV-associated HCC risk prediction and immunotherapy [125].

### 4.9. Herpesvirus

Herpesviruses are double-stranded DNA viruses belonging to the *Herpesviridae* family [126]. Their life cycles usually can be divided into latency and lytic reactivation. Among them, Kaposi’s sarcoma-associated herpesvirus (KSHV) and Epstein-Barr virus (EBV) are further categorized as gammaherpesviruses. KSHV infection induces Kaposi’s sarcoma (KS), primary effusion lymphoma (PEL), and multicentric Castleman’s disease (MCD), while EBV infection has been widely related to nasopharyngeal carcinoma (NPC), gastric carcinoma (GC), and lymphoma [127,128]. Human cytomegalovirus (HCMV) belongs to betaherpesviruses and poses a health risk to immunocompromised individuals [129]. In addition, herpes simplex virus type 1 (HSV-1), a member of alphaherpesviruses, can cause many human diseases, including genital herpes, cold sores, herpes stromal keratitis, encephalitis, and meningitis [130].

The m6A modifications are predominantly deposited in the GGAC motifs of KSHV transcripts during latent infection and lytic replication [131,132,133,134]. The distribution of m6A modifications in viral transcripts varies among different infected cell types, though there are partially overlapping modification peaks [132,133,134]. A significantly increased m6A level is present on viral mRNAs during viral lytic replication, compared to the latent state [131,132,135]. In addition, knockdown of METTL3 or YTHDF2 significantly suppressed the expression of viral lytic genes and the production of viral particles, while knockdown of FTO showed a beneficial effect, emphasizing the importance of m6A modifications in the KSHV lytic replication cycle [131,133,135]. In another study, YTHDF2 was found to mediate the destabilization of viral lytic transcripts, which reduced KSHV lytic replication [132]. Of interest, viral replication transcriptional activator ORF50 (RTA) RNA is m6A-modified in different cell types. After maturation by splicing, ORF50 pre-mRNAs express RTA proteins, which assist in the transition from latent to lytic replication [131,133,135]. Three m6A sites in the ORF50 intron (A71926, A71944, A72572) and one (A72717) in ORF50 exon 2 are essential for ORF50 pre-mRNA splicing and RTA protein expression. In particular, the m6A modifications located in the ORF50 intron aid RTA pre-mRNA splicing by recruiting YTHDC1 and serine/arginine-rich splicing factors SRSF3 and SRSF10. Furthermore, RTA expression increases m6A levels in ORF50 mRNA to enhance its own pre-mRNA splicing [131]. Baquero-Perez et al. identified a novel m6A reader, staphylococcal nuclease domain-containing protein 1 (SND1), which was found to maintain ORF50 RNA stability and thus aid viral lytic replication. METTL3-mediated m6A modifications in ORF50 RNA are beneficial for SND1 recruitment to ORF50 RNA [133]. Overall, m6A modifications in KSHV transcripts contribute to viral lytic replication.

Extensive m6A modifications are known to occur on EBV transcripts across different life cycles and cell types, and they are predominantly dispersed in the CDS region of viral mRNAs [136,137,138]. However, unlike KSHV, more m6A modifications are deposited in EBV transcripts during latent infection. METTL14, METTL3, or YTHDF1 positively regulate the stability and expression of latent gene transcripts, while negatively regulating viral lytic replication and viral particle production [136,137,138]. In addition, YTHDF1 and YTHDF2 promote the decay of viral BZLF1 and BRLF1 transcripts to inhibit viral lysis reactivation, a process in which m6A in BZLF1 and BRLF1 mRNAs may be an important prerequisite for the recruitment of these m6A readers [137,139]. The m6A modifications in viral latent gene *EBNA3C* transcript promote the EBNA3C mRNA stability and expression level, and EBV latent infection or potential antigen EBNA3C expression can further mediate a positive feedback on the EBNA3C mRNA level by promoting METTL14 transcription and protein stability [136]. In general, m6A modifications may help maintain latent EBV infection while inhibiting viral lytic replication, though one study reported ambiguous results [140]. More importantly, m6A regulators, including METTL3, METTL14, WTAP, FTO, and IGFBP1, regulate the proliferation of multiple EBV-associated carcinoma cells and tumor growth in vivo, suggesting that these regulators can serve as therapeutic targets for EBV-associated cancers [141,142,143,144].

Some m6A modifications are present in the three HCMV long non-coding RNAs (lncRNAs), RNA1.2, RNA2.7, and RNA4.9, which are necessary in the HCMV life cycle [129]. YTHDF2 and IGF2BP3 interact with these lncRNAs and promote their stability, but the detailed mechanism remains to be determined [129]. In addition, METTL14 and ALKBH5 regulate viral innate immune responses and viral replication [145].

HSV-1 transcripts contain many m6A modifications [146,147]. Although the function of m6A sites in HSV-1 transcripts has been barely elucidated, there are several pieces of evidence supporting the idea that m6A modifications on viral or cellular RNAs potentially affect viral replication and antiviral immunity [147,148,149,150]. For instance, METTL14 can enhance ISG15 mRNA stability and ISG15 expression by increasing the m6A level of ISG15 mRNA, and therefore promoting anti-HSV-1 effects. Notably, the virus-encoded immediate-early protein ICP0 causes METTL14 ubiquitination and proteasomal degradation during early HSV-1 infection [148]. Additionally, Li et al. have found that enhanced ALKBH5 lactylation at lysine K284 during herpesvirus infections mediates the m6A demethylation of interferon-beta (IFN-β) mRNA and facilitates IFN-β mRNA biogenesis, which in turn contributes to the antiviral innate immunity [150].

### 4.10. SV40

Simian virus 40 (SV40) belongs to the *Polyomaviridae* family, and has a circular double-stranded DNA genome which is divided into an early region encoding viral regulatory proteins and a late region encoding viral structural proteins [151]. SV40 can induce malignant transformation of some normal human and animal cells in vitro. The connection between SV40 and human tumors has been extensively studied, but remains highly controversial [152,153].

Early studies reported the presence of m6A modifications in the coding region of SV40 transcripts, but did not clarify their specific modification sites and functions [66,68]. In 2018, Tsai et al. mapped two m6A peaks in SV40 early transcripts and eleven m6A peaks in late transcripts by PA-m6A-seq, which coincided with the YTHDF2 and YTHDF3 binding sites. Among them, m6A modifications in the late transcripts are mainly distributed in the open reading frame (ORF) of structural protein VP1 and promote VP1 mRNA translation and viral replication, without affecting viral late mRNA splicing. In addition, YTHDF2 overexpression promoted the expression of VP1 and viral regulatory protein TAg as well as phagolysin formation, whereas YTHDF2 or METTL3 knockdown inhibited SV40 replication [154]. In conclusion, m6A has a positive role in regulating SV40 gene expression and viral replication.

### 4.11. HAdV

Human adenoviruses (HAdVs) belong to the genus *Mastadenovirus*, and have a linear, double-stranded DNA genome. HAdV infections are associated with acute febrile respiratory diseases and a variety of inflammatory diseases [155].

Some m6A modifications are present on both early and late HAdV transcripts. The m6A modifications in late transcripts, such as Fiber pre-mRNA, are important for their splicing. METTL3 favors the increased splicing efficiency of late transcripts to promote their gene expression and virion production without affecting early viral transcripts [156]. In addition, Hajikhezri et al. found that m6A levels in viral major late transcription unit (MLTU) mRNAs, including pVII and Fiber transcripts, were positively correlated with the expression of fragile X mental retardation protein 1 (FXR1) isoforms. These m6A modifications may help FXR1 bind to MLTU mRNAs to exert a role for FXR1 in the HAdV life cycle [157].

**Table 1 microorganisms-12-02373-t001:** The m6A modifications in viral RNAs and their roles.

Virus	Site on Viral RNA	Function	Reference
HIV-1 (ssRNA-RT)	RRE RNA (A7883)	Promotes viral RNA nuclear export and viral replication.	[70]
	3′ UTR	Promotes viral mRNA expression.	[71]
	5′ UTR	Promotes viral infectivity.	[74]
	Viral RNAs	Inhibits IFN-I induction.	[77]
SARS-CoV-2 (+ssRNA)	*N* region	Helps evade RIG-I-mediated innate immunity.	[37]
	A74 in SL3 hairpin/TRS-L of 5′ UTR	Reduces the stability of 5′ SL3 and TRS RNA duplexes and affects discontinuous transcription of four structural protein sgRNAs.	[84]
IAV (−ssRNA)	HA mRNA and vRNA	Promotes HA mRNA levels and protein expression.	[43]
	3′ splicing site of NS mRNA (A530)	Inhibits viral NS mRNA splicing to promote NS1 expression and facilitate IAV replication and pathogenicity.	[88]
HCV (+ssRNA)	*E1* region	Inhibits assembly and release of infectious virus particles.	[91]
	PAMP RNA upstream (A8766)	Helps evade RIG-I-mediated I-IFN signaling response.	[95]
	IRES element (A331)	Enhances HCV IRES-dependent translation initiation.	[92]
ZIKA (+ssRNA)	Viral RNAs	May affect viral replication.	[36,94]
DENV, YFV, and WNV (+ssRNA)	Viral gRNAs	Uncertain.	[91]
EV71 (+ssRNA)	A3055 in *VP1* region and A4555 in *2C* region	Promotes progeny virus production.	[97]
	5′ UTR-*VP4* junction	Facilitates viral translation.	[98]
hMPV (−ssRNA)	*G* gene region	Helps evade RIG-I-mediated innate immunity.	[105]
RSV (−ssRNA)	*G* gene region	Helps evade RIG-I-mediated innate immunity.	[106]
VSV (−ssRNA)	Viral transcripts	Helps evade RIG-I-mediated innate immunity.	[110]
HBV (dsDNA-RT)	5′ and 3′ epsilon stem loop of pgRNA (A1907)	The m6A1907 in the 3′ epsilon stem loop decreases HBV RNA stability and viral protein expression, whereas m6A1907 in the 5′ epsilon stem loop helps the nuclear export of HBV transcripts, reverse transcription and cccDNA synthesis.	[115,116,117]
	5′ and 3′ epsilon stem loop of pgRNA (A1907)	Supports ISG20-mediated viral RNA degradation.	[119]
	5′ epsilon stem loop of pgRNA (A1907)	Inhibits RIG-I recognition and phosphorylation of IRF3.	[95]
	3′ UTR of HBx coding region (A1907)	Promotes HBx mRNA decay.	[122]
	HBx coding region (A1616)	Specifically inhibits HBx mRNA and protein levels.	[118]
	C-terminal of HBx coding region (A1662, A1670, A1714 and A1729)	Promotes HBV mRNA and HBs protein levels.	[120]
KSHV (dsDNA)	ORF50 (RTA)	Helps ORF50 pre-mRNA splicing and RTA protein expression.	[131]
EBV (dsDNA)	EBNA3C transcript	Promotes the expression level and stability of EBNA3C mRNA.	[136]
HCMV (dsDNA)	Viral lncRNAs	May promote the lncRNAs’ stability.	[129]
HSV-1 (dsDNA)	Viral transcripts	Uncertain.	[147]
SV40 (dsDNA)	Viral late transcripts	Promotes translation of late transcripts and viral replication.	[154]
HAdV (dsDNA)	Viral late transcripts	Promotes the splicing efficiency of late transcripts	[156]

Note: single-stranded RNA reverse transcribing (ssRNA-RT), positive-sense single-stranded RNA (+ssRNA), negative-sense single-stranded RNA (−ssRNA), double-stranded DNA reverse transcribing (dsDNA-RT), and double-stranded DNA (dsDNA).

## 5. Roles of m5C in Viral RNAs

Recently, the existence and function of m5C in some viral RNAs has been characterized. Herein, we summarize the typical mechanisms of m5C modifications in viral RNAs regulating viral infection (Table 2).

### 5.1. HIV-1

Courtney et al. quantified 11 m5C-modified residues in HIV-1 gRNA by UPLC-MS/MS and mapped the m5C modification profile by PA-m5C-seq. The NSUN2 protein is responsible for the addition of most of these modifications [44]. In addition, Cristinelli et al. confirmed that m5C modifications do exist in HIV RNAs by BS-Seq [158], but another study using the same method did not detect the m5C methylation in HIV-1 RNAs, probably due to different experimental conditions and analytical procedures [159]. The m5C writers NSUN1 and NSUN2 regulate different life-cycle stages of HIV-1. NSUN1 competitively binds to the HIV-1 transactivation response (TAR) element and mediates methylation of its cytosine through the RNA MTase catalytic structural domain, which reduces the interaction of the Tat protein with TAR RNA, and in turn disturbs viral transcription and lytic replication, but does not clarify m5C modification’s direct function [160]. NSUN2 does not affect viral transcription but facilitates *gag* mRNA binding to the ribosome, which is dependent on m5C modification in viral mRNAs, suggesting that NSUN2-mediated m5C modification in viral transcripts may play an important role in viral translation. In addition, NSUN2-catalyzed m5C modifications, which are deposited in the overlapping region of viral *pol* 3′ end and *vif* 5′ end and close to the HIV-1 *A2* splice site, can promote the selective splicing of viral RNAs [44].

### 5.2. SARS-CoV-2

LC-MS/MS, m5C-MeRIP-seq, and BS-seq data have shown that SARS-CoV-2 gRNAs carry abundant m5C modifications [40,161]. However, Huang et al. did not find the m5C methylation in SARS-CoV-2 RNAs using BS-seq [159]. Wang et al. further explored the effect of m5C modification on the life cycle of SARS-CoV-2 [40]. They found that low levels of m5C modification in viral RNAs caused by NSUN2 depletion were beneficial to viral replication and infection. In addition, NSUN2-mediated m5C modification in different regions of viral transcripts, including nsp14, nsp15, nsp16, S, E, M, and N, reduced the stability and levels of corresponding mRNA. More importantly, NSUN2 knockout mice showed more severe SARS-CoV-2 virus infection and lung tissue damage compared to control mice. The progeny SARS-CoV-2 particles with low m5C modifications produced by these mice also showed enhanced replication and virulence in vivo. Clinically, low expression of NSUN2 is characterized in bronchoalveolar lavage fluid (BALF) from severe COVID-19 patients. In summary, m5C modifications added by NSUN2 in SARS-CoV-2 transcripts are detrimental to viral replication and pathogenicity.

### 5.3. HCV

Li et al. have identified a crucial m5C site for the HCV life cycle by use of BS-seq analysis, one which is located at C7525 in the *NS5A* region of the HCV genome [51]. Mutation at this site promoted the decay of HCV RNA and inhibited HCV RNA replication, protein expression, and viral release. In addition, the m5C reader YBX1 bound to HCV RNA by recognizing the m5C7525 locus to help HCV replication. Importantly, YBX1 depletion in the liver of humanized transgenic mice significantly inhibited HCV replication and reduced tissue damage and inflammatory infiltration in vivo, highlighting the pro-viral effect of m5C and its related cellular proteins, which can be potential antiviral therapeutic targets.

### 5.4. EV71

The m5C modifications are mainly located in the 5′ UTR and the CDS regions, including those of protease 3C, polymerase 3D, and four structural proteins (VP1-4) of the EV71 genome [58]. Functionally, an m5C site (nt 584) located within the IRES motif of 5′ UTR can promote EV71 translation and RNA stability, while another m5C site (nt 1460), located in the *VP2* CDS region, also facilitates the stability of viral RNAs. NSUN2 is essential for the functioning of these m5C sites. Moreover, mutation at C584 or C1460 can inhibit EV71 replication and progeny virus production in vitro, and attenuate viral pathogenicity in vivo, emphasizing the anti-EV71 effect of m5C modifications.

### 5.5. HBV

A significant amount of m5C modification catalyzed by NSUN2 has been reported in HBV RNA, a phenomenon which has positive regulatory effects on the viral life cycle [49,52,162]. In detail, the m5C modification located at the conserved site C2017 of the viral HBc protein ORF contributes to viral RNA stability and viral replication [162]. Four m5C sites identified in HBV mRNAs by Ding et al. are critical for viral replication and viral antigen secretion. In particular, methylation at C1291 can be recognized by ALYREF to promote nuclear export of HBV mRNAs. Furthermore, this m5C modification also enhances mRNA translation and helps viral RNA evade RIG-I recognition [52]. The m5C modifications in the HBV 5′ terminal epsilon hairpin are critical for viral pgRNA packaging and reverse transcription. In addition, they also have promoting effects on levels of HBV capsid protein HBc and genomic DNA in secreted progeny virus particles [49]. Consistent with this, NSUN2 positively regulates HBV replication and progeny viral particle production, whereas the m5C eraser TET2 exhibits negative regulation of viral replication [49,162]. Nonetheless, a contrasting report suggests that NSUN2 might exert a suppressive effect on HBV replication by enhancing IFN production through the catalysis of m5C modifications in I-IFN RNA [52]. In summary, m5C modifications in HBV RNA primarily play a facilitating role in viral replication.

### 5.6. EBV

EBV-expressed noncoding RNA EBER1 is m5C-modified at C145, which is catalyzed by NSUN2. NSUN2 knockout completely eliminated m5C methylation at C145 and increased EBER1 RNA levels, possibly because m5C modification at this site facilitated targeted degradation of EBER1 by RNase ANG. However, EBV lytic replication was not affected by NSUN2 deletion [163]. Whether m5C modifications in EBV RNAs regulate the viral life cycle needs to be further explored.

**Table 2 microorganisms-12-02373-t002:** The m5C modifications in viral RNAs and their roles.

Virus	Site on Viral RNA	Function	Reference
HIV-1 (ssRNA-RT)	Overlap of *pol* 3′ end and *vif* 5′ end	Promotes selective splicing of viral RNA.	[44]
SARS-CoV-2 (+ssRNA)	Viral gRNA	Promotes degradation of viral transcripts and inhibits viral replication and pathogenicity.	[40]
HCV (+ssRNA)	*NS5A* region (C7525)	Promotes HCV RNA stability, viral replication and viral progeny release.	[51]
EV71 (+ssRNA)	IRES of 5′ UTR (C584) and *VP2* CDS region (C1460)	Promotes viral translation and RNA stability, favoring viral replication and pathogenicity.	[58]
HBV (dsDNA-RT)	HBc ORF (C2017)	Contributes to pgRNA stability and promotes viral replication, both in vitro and in vivo.	[162]
	Viral mRNA (C1291)	Promotes nuclear export and translation of HBV mRNA and helps viral RNA evade RIG-I recognition.	[52]
	5′ epsilon hairpin (C1842, C1845, C1847, C1858, and C1859)	Promotes viral pgRNA packaging and the reverse transcription process.	[49]
EBV (dsDNA)	EBER1 RNA (C145)	May decrease the EBER1 stability.	[163]

Note: single-stranded RNA reverse transcribing (ssRNA-RT), positive-sense single-stranded RNA (+ssRNA), negative-sense single-stranded RNA (−ssRNA), double-stranded DNA reverse transcribing (dsDNA-RT), and double-stranded DNA (dsDNA).

## 6. Roles of ac4C in Viral RNAs

Since 2020, the effects of ac4C modification on viral replication have been gradually elucidated in some viruses, including HIV-1, EV71, and KSHV; this section provides a summary of these findings. Additionally, ac4C modifications are also deposited in the negative-strand RNA of IAV, yet the precise roles of this modification are still uncharted territories (Table 3) [42].

### 6.1. HIV-1

In 2020, Tsai et al. concretely mapped the ac4C modification profile in HIV-1 transcripts by PA-ac4C-seq and PAR-CLIP-seq, identifying a total of 11 sites modified by NAT10 [45]. RNA acetyltransferase activity of NAT10 is important for HIV-1 replication. The ac4C modifications on viral transcripts positively regulate HIV-1 gene expression and replication, which can be explained by enhanced stability of HIV-1 transcripts being achieved without affecting viral translation efficiency [45].

### 6.2. EV71

NAT10 mediates acetylation of the 5′ UTR in EV71 mRNA, with particular ac4C modifications identified in stem-loop IV of the IRES, specifically, at two confirmed positions, C331 and C350 [41]. These ac4C modifications contribute to the stability of viral RNA and promote the initiation of EV71 protein synthesis by selectively enhancing the binding of the IRES trans-acting factor poly(rC)-binding protein 2 (PCBP2) to IRES. Furthermore, the two ac4C sites strengthen the interaction between viral RdRp (3D) and EV71 RNA, thereby promoting viral RNA replication. Overall, ac4C modifications on the EV71 5′ UTR have a promotional function on EV71 replication. Most notably, the ac4C-mutated viral strain exhibited diminished replication capabilities in mice and induced attenuated pathological damage, implying that targeting the ac4C modification in EV71 RNAs could be a viable strategy for antiviral intervention.

### 6.3. KSHV

The most abundant ac4C modification peaks were detected on KSHV polyadenylated nuclear RNA (PAN), a virus-encoded long noncoding RNA. NAT10, along with its catalytic ac4C modification in KSHV PAN RNA, can enhance the stability and expression of PAN RNA, thereby facilitating the lytic replication of KSHV. Additionally, ac4C modifications in PAN RNA enhance the binding affinity for NAT10 to IFN-γ-inducible protein-16 (IFI16) mRNA, resulting in an elevated cytidine acetylation level in IFI16 mRNA and promoting the expression of IFI16 protein as well as inflammatory vesicle activation [53]. Moreover, ac4C modification on tRNA during KSHV replication promotes viral lytic gene translation and viral particle production [164]. To sum up, ac4C is critical for the reactivation of KSHV from latency and antiviral immunity.

## 7. Roles of m1A in Viral RNAs

Some m1A modifications have been detected in IAV negative-strand PB2, HA, and M RNAs, with unknown effects [42]. However, m1A modifications at the A58 position in the host specific tRNA primer, which is necessary for the reverse transcription of retroviruses including HIV-1 and human T-cell leukemia virus type 1 (HTLV-1), play a pivotal role in terminating the synthesis of viral plus-stranded DNA and facilitating plus-strand transfer [165,166,167]. In addition, mutations at the m1A site in tRNA primers inhibited HIV-1 replication and infection [166,168]. Knockdown of the m1A writer TRMT61A inhibited progeny particle production of influenza virus [42]. Thus, there may be a pro-viral effect of m1A modification on retroviruses and influenza viruses. However, the m1A modification appeared to repress the replication capacity of SARS-CoV-2, because in vitro experiments showed that m1A inhibited the RNA elongation activity of SARS-CoV-2 replication complex (SC2RC) [169]. More evidence is still needed in the future to explain the role of m1A modification in viral RNAs.

## 8. Application of RNA Modifications in Antiviral Therapeutics and Vaccine Development

As intracellular parasitic organisms, most viruses share the same machinery with their host cells in RNA modification, including writers, erasers, and readers. To date, the only exception found is that SND1 serves as a novel m6A reader, one which is recruited by m6A-modified KSHV RNA [133]. The m6A/m5C/ac4C machinery proteins have been found to be key regulators of viral replication and pathogenicity by their mediating of the addition, removal, and recognition of viral RNA modifications (Figure 1). In addition, viral infection and viral proteins can alter the subcellular localizations and expression patterns of RNA modification machinery proteins. This alteration further promotes modification-mediated pro-viral effects, or inhibits anti-viral effects, in order to support viral self-infection and replication [40,41,51,57,58,76,146,148,162]. Consequently, the modulation of RNA modification machinery proteins represents a significant avenue for developing antiviral strategies.

Promisingly, many small-molecule compounds targeting RNA modifications have already been tested for antiviral efficacy in cellular or animal models. 3-deazaadenosine (DAA) inhibits the production of the methyl donor S-adenosylmethionine (SAM) by depressing S-adenosylhomocysteine (SAH) hydrolase to reduce the addition of m6A and m5C in RNAs [170,171]. As a nonspecific inhibitor of m6A, DAA can effectively suppress the replication of RNA viruses, such as HIV-1 [71], IAV [43], and EV71 [98], as well as DNA viruses like KSHV [131], EBV [140], and HSV-1 [147]. For example, DAA suppresses KSHV RTA precursor mRNA splicing and lytic gene expression while inhibiting m6A levels [131]. Although the specific mechanism of the antiviral activity of DAA has not been fully clarified, DAA is still a promising candidate for broad-spectrum antiviral drugs. In addition, Kumar et al. reported the anti-SARS-CoV-2 activity of 3-Deazaneplanocin A (DZNep) [19]. DZNep significantly reduces the m6A level on viral RNA during SARS-CoV-2 infection, and inhibits viral RNA and protein synthesis by blocking the interactions of viral RNA with the m6A reader hnRNPA1 and eukaryotic translation initiation factor eIF4E. Importantly, DZNep is less likely to induce drug resistance in viruses and may exhibit broad-spectrum anti-coronavirus effects [19]. Similarly, cycloleucine, an m6A methylation inhibitor, can reduce HBV RNA levels and protein production [120]; additionally, the METTL3 inhibitor STM2457 has been found to suppress the replication and production of viruses by inhibiting the early synthesis of human betacoronavirus RNA and nucleocapsid (N) protein [80]. In addition to m6A inhibitors, 5-azacytidine (5-AzaC), which disrupts m5C methyl transfer, can hinder HBV replication and viral progeny secretion [49]. Remodelin, an inhibitor of NAT10 activity, can be used to restrict HIV-1 and EV71 replication [41,45,172].

Many natural compounds have also been reported to interfere with viral replication via their targeting of RNA modification machinery. Rhein (4,5-dihydroxyanthraquinone-2-carboxylic acid) extracted from traditional medicinal plants inhibits FTO catalytic activity and exhibits antiviral effects on three human coronaviruses infections [173]. Schisandra chinensis, another traditional Chinese medicine derived from plants, has also been found to promote the m6A levels in the coding region of bovine herpesvirus-1 (BoHV-1) envelope glycoprotein gD, which may inhibit gD transcription and thereby hinder BoHV-1 replication [174]. 

In summary, targeting RNA modification processes is an important direction for the screening and development of pan-antiviral agents. Nevertheless, most identified drugs that affect RNA modification machineries have only been validated for antiviral activity in vitro, and more in vivo evidence remains to be provided. In addition, the antiviral mechanisms of these drugs need to be further elucidated.

Viral RNA modifications also provide a promising strategy for vaccine design. For instance, the m6A-deficient RSV and hMPV strains obtained by synonymously mutating the m6A sites in the viral *G* gene show lower pathogenicity but retain high immunogenicity in cotton rats, suggesting that the removal of functional m6A sites in viral RNA can be used to construct live attenuated vaccines [104,105,106]. On the other hand, incorporating m6A and pseudouridine modifications into the conserved *HA* region, as well as the *M2* region, in influenza viruses may help the stability and efficacy of mRNA vaccines in vivo [175]. Although these studies are still within the area’s infancy, we can expect that viral RNA modifications will provide more targets and tools for vaccine development. 

## 9. Discussion and Conclusions

RNA modifications are a newly recognized epigenomic regulator critical for gene expression and phenotypic determination. Compared to cells, the life cycles of viruses depend much more on RNA. For RNA viruses, RNAs serve as both genomes and transcripts, constituting most stages of their life cycle; even for DNA viruses, posttranscriptional modifications on RNA may lead to significant changes, due to their compact genome with low redundancy. Especially for pathogenic viruses, viral RNA modifications may affect the development of diseases, and thus can serve as drug targets. Although the field has been teeming with studies of viral RNA modifications in recent years, the overall knowledge of them is still insufficient and needs to be better organized. Herein, we systematically summarize the current progression of the research on RNA modification in pathogenic viruses, aiming to providing a comprehensive reference for advanced studies in this field.

Similar to cellular RNA modifications, m6A is the most prevalent and extensively studied modification across various DNA and RNA viruses. In contrast, m5C and ac4C have been only identified in a subset of viral RNAs as serving distinct roles. Although the specific functions of viral m1A modification have not yet been reported, the presence of this modification in RNA viruses suggests that it may have important biological significance. At present, the functional m6A, m5C, and ac4C sites are primarily found in viral transcripts, with a smaller proportion being detected in viral negative-stranded RNAs and non-coding RNAs. These modifications can enhance or repress viral replication and infectivity by affecting the stability, splicing, nuclear export, packaging, reverse transcription, and translation of viral RNAs. The specific functions of certain modifications mainly depend on the viral species and the location where RNA modifications are deposited. Furthermore, m6A and m5C have been identified as key mechanisms for various viruses in their escape from the RIG-I-dependent innate immune response.

In conclusion, viral RNA modifications play important regulatory roles in various viral life cycles, revealing a series of novel mechanisms of viral infection, variation, and virus–host interactions, greatly enriching the knowledge of epigenomics. Especially, research on RNA modifications of pathogenic viruses not only refreshes our understanding of viral diseases, but also provides new strategies and targets for the development of antiviral vaccines and therapeutics. Most recent research in this field tends to focus on the conserved modification sites among different viruses and their functional mechanisms. Notably, the exploration of viral RNA modification is promising, but still at the very beginning stage. More techniques for RNA modification detection with high sensitivity, high accuracy, low cost, and single-base resolution are required to support further research on viral RNA modification.

## Figures and Tables

**Figure 1 microorganisms-12-02373-f001:**
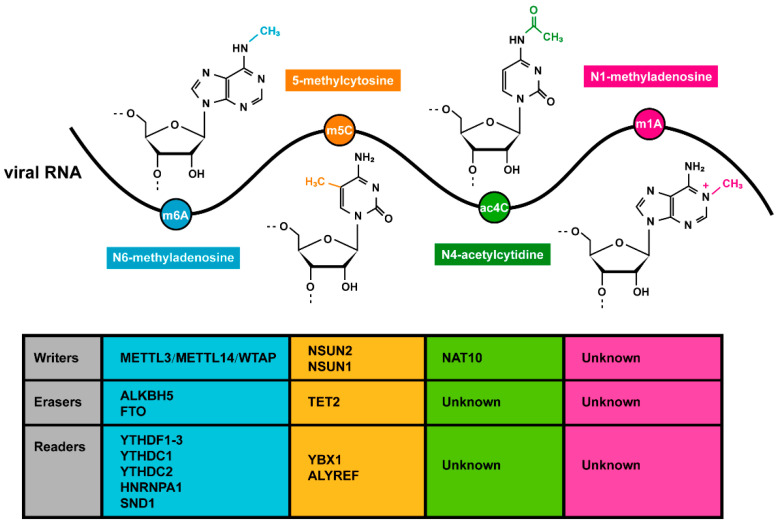
Viral RNA modifications. Chemical structures of four RNA modifications deposited in viral RNAs are shown, including N6-methyladenosine(m6A), 5-methylcytosine(m5C), N4-acetylcytidine(ac4C), and N1-methyladenosine(m1A). Host proteins responsible for the addition, removal, or recognition of these modifications in viral RNAs are listed.

**Table 3 microorganisms-12-02373-t003:** The ac4C modifications in viral RNAs and their roles.

Virus	Site on Viral RNA	Function	Reference
HIV-1 (ssRNA-RT)	Viral transcripts (*env* CDS)	Promotes HIV-1 gene expression and replication, primarily by enhancing viral RNA stability.	[45]
EV71 (+ssRNA)	IRES of 5′ UTR in EV71 mRNA (C331 and C350)	Promotes viral mRNA stability and translation initiation.	[41]
KSHV (dsDNA)	PAN RNA	Promotes viral lytic replication.	[53]
IAV ((-(−ssRNA)	Viral negative-sense RNAs	Uncertain.	[42]

Note: single-stranded RNA reverse transcribing (ssRNA-RT), positive-sense single-stranded RNA (+ssRNA), negative-sense single-stranded RNA (−ssRNA), double-stranded DNA reverse transcribing (dsDNA-RT), and double-stranded DNA (dsDNA).

## Data Availability

No new data were created or analyzed in this study. Data sharing is not applicable to this article.

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
