# Peer review of "RNA Modifications in Pathogenic Viruses: Existence, Mechanism, and Impacts"

_microorganisms, 2024, doi:10.3390/microorganisms12112373_

Round 1
Reviewer 1 Report
Comments and Suggestions for Authors
The present review article "RNA Modifications in Pathogenic Virus: Existence, Mechanism and Impacts" describes the variety of RNA modifications and their influence in different stages of viral life cycle. Different human viruses have been analysed, including HIV, herpesvirus, hepatitis virus, coronavirus, giving a very fine view of these type of modifications during the infection. The article is well written and finely organized, resulting suitable for acceptance after minor revisions essentially directed to update the reference section. Please take into consideration: doi: 10.3390/ph14111135; doi: 10.3389/fmicb.2024.1444414; doi: 10.1007/s13577-024-01044-3; doi: 10.1073/pnas.2409132121.
Author Response
Dear reviewer,
Thank you very much for your valuable comments and suggestions on this manuscript! We have revised the manuscript accordingly. Please see the following response to the comment.
Comment: The present review article "RNA Modifications in Pathogenic Virus: Existence, Mechanism and Impacts" describes the variety of RNA modifications and their influence in different stages of viral life cycle. Different human viruses have been analysed, including HIV, herpesvirus, hepatitis virus, coronavirus, giving a very fine view of these type of modifications during the infection. The article is well written and finely organized, resulting suitable for acceptance after minor revisions essentially directed to update the reference section. Please take into consideration: doi: 10.3390/ph14111135; doi: 10.3389/fmicb.2024.1444414; doi: 10.1007/s13577-024-01044-3; doi: 10.1073/pnas.2409132121.
Response: Thank you for the detailed comments and the valuable suggestions providing the related references we have not covered. In the revision, we have added the recommended references (citations 125, 150, 173 and 174) to the relevant sections and described their content briefly, which are highlighted in yellow. We hope the revision could improve the quality of our manuscript.
Reviewer 2 Report
Comments and Suggestions for Authors
This is a very well written and extensive review of modifications of viral RNAs. The topic is highly relevant for the understanding of viral pathology and the development of new antivirals.
My criticism is minor:
The authors list various methods for the detection of viral RNA modifications (introduction); it would be beneficial to add information about the advantages and pitfalls associated with each. Perhaps a subtitle (e.g. Methods of RNA Modifications detection) would be warranted.
The part on antivirals would benefit from being put under a subtitle and should be more detailed as this is of major interest.
Other:
Abbreviations should be expanded in full when first appearing. This is not uniform
‘neurological infiltration’ – neuroinflammation would be a more appropriate term
‘SV40 is linked to a variety of human malignancies’ – this is an overstatement as the role of this virus in human pathology remains highly controversial.
Author Response
Dear reviewer,
Thank you very much for your constructive comments and valuable suggestions on this manuscript! We have carefully revised the manuscript and provided the point-by-point response below. The changes in the revised manuscript have been marked in red. Your time and effort are greatly appreciated. If you have any questions, please do not hesitate to let us know.
|
Comment 1: The authors list various methods for the detection of viral RNA modifications (introduction); it would be beneficial to add information about the advantages and pitfalls associated with each. Perhaps a subtitle (e.g. Methods of RNA Modifications detection) would be warranted. Response 1: Thank you for the constructive suggestions! In the revised manuscript, we have summarized current methods for viral RNA modification detection and evaluated their advantages and pitfalls in an individual section with the subtitle “3. Methods of viral RNA modifications detection”. We hope that our revision (marked in red) will improve the coverage and the readability of our manuscript. |
|
Comment 2: The part on antivirals would benefit from being put under a subtitle and should be more detailed as this is of major interest. Response 2: Thank you for the constructive suggestions! In the revised manuscript, we describe the potential of viral RNA modifications as the targets for antiviral and vaccine development in a new section with the subtitle “8. Application of RNA modifications in antiviral therapeutics and vaccine development”. Especially, we have added some details about the antiviral mechanism of chemical drugs and natural components that are associated with RNA modification. We sincerely hope that the revised content in this section, which has been marked in red, can improve the quality of our manuscript. |
|
Comment 3: Abbreviations should be expanded in full when first appearing. This is not uniform ‘neurological infiltration’ – neuroinflammation would be a more appropriate term. Response 3: Thank you for your careful reading and critical comments! We have proofread the whole manuscript and added the full terms for all abbreviations where they first appeared, which are all marked in red. Also, we have replaced all the “neurological infiltration” with “neuroinflammation” according to the suggestion. |
|
Comment 4: ‘SV40 is linked to a variety of human malignancies’ – this is an overstatement as the role of this virus in human pathology remains highly controversial. Response 4: Thank you very much for the critical comment! With further literature review, we agree that this sentence is an overstatement, considering the role of SV40 infection in human oncogenesis is still controversial. We have weakened the statement to “SV40 can induce malignant transformation of some human and animal normal cells in vitro. The connection between SV40 and human tumors has been extensively studied, but remains highly controversial [146, 147].” We sincerely hope that the revised expression is more appropriate. |